# Nasal Microbiota Modifies the Effects of Particulate Air Pollution on Plasma Extracellular Vesicles

**DOI:** 10.3390/ijerph17020611

**Published:** 2020-01-17

**Authors:** Jacopo Mariani, Chiara Favero, Michele Carugno, Laura Pergoli, Luca Ferrari, Matteo Bonzini, Andrea Cattaneo, Angela Cecilia Pesatori, Valentina Bollati

**Affiliations:** 1EPIGET LAB, Department of Clinical Sciences and Community Health, Università degli Studi di Milano, 20122 Milan, Italy; jacopo.mariani@unimi.it (J.M.); chiara.favero@unimi.it (C.F.); michele.carugno@unimi.it (M.C.); laura.pergoli@unimi.it (L.P.); luca.ferrari@unimi.it (L.F.); matteo.bonzini@unimi.it (M.B.); angela.pesatori@unimi.it (A.C.P.); 2Fondazione IRCCS Ca’ Granda Ospedale Maggiore Policlinico, 20122 Milan, Italy; 3Department of Science and High Technology, University of Insubria, 22100 Como, Italy; andrea.cattaneo@uninsubria.it

**Keywords:** particulate matter, extracellular vesicles, nanoparticle tracking analysis, flow-cytometry, dysbiosis, bacteria, *Moraxella*, nasal microbiota, microbiome, 16S

## Abstract

Air pollution exposure has been linked to modifications of both extracellular vesicle (EV) concentration and nasal microbiota structure (NMB), which might act as the respiratory health gatekeeper. This study aimed to assess whether an unbalanced NMB could modify the effect of particulate matter (PM) exposure on plasmatic EV levels. Due to two different NMB taxonomical profiles characterized by a widely different relative abundance of the *Moraxella* genus, the enrolled population was stratified into Mor− (balanced NMB) and Mor+ (unbalanced NMB) groups (*Moraxella* genus’s cut-off ≤25% and >25%, respectively). EV features were assessed by nanoparticle tracking analysis (NTA) and flow-cytometry (FC). Multivariable analyses were applied on EV outcomes to evaluate a possible association between PM10 and PM2.5 and plasmatic EV levels. The Mor− group revealed positive associations between PM levels and plasmatic CD105+ EVs (GMR = 4.39 *p* = 0.02) as for total EV count (GMR = 1.92 *p* = 0.02). Conversely, the Mor+ group showed a negative association between exposure and EV outcomes (CD66+ GMR = 0.004 *p* = 0.01; EpCAM+ GMR = 0.005 *p* = 0.01). Our findings provide an insight regarding how a balanced NMB may help to counteract PM exposure effects in terms of plasmatic EV concentration. Further research is necessary to understand the relationship between the host and the NMB to disentangle the mechanism exerted by inhaled pollutants in modulating EVs and NMB.

## 1. Introduction

Extracellular vesicles (EVs) are powerful and not yet fully understood biological effectors shared between domains of life [1].

Several biological molecules have been identified to be carried into the EVs, such as DNA, small RNA, and non-coding RNA, including also miRNAs of different size [2,3], proteins, and other soluble factors [4], which are internalized by recipient cells after either EV interaction through surface-expressed ligands or endocytosis [5]. Different studies have underlined that EVs are involved in numerous biological and pathological processes, which span from immune system modulation [6,7], cancer [8], metabolic diseases [9], atherosclerosis [10], and development of chronic obstructive pulmonary disease (COPD) to allergic airway inflammation [11]. It has been found that EV production could be influenced and involved in response to volatile pollutant exposure, including particulate matter (PM) [12,13].

PM is defined as the heterogeneous mixture of both organic and non-organic particles, which derives from several sources and generally is sorted due to particle aerodynamic diameter into PM10 and PM2.5 (diameter ≤10 µm or ≤2.5 µm, respectively). According to the World Health Organization, exposure to PM has been linked to an increased morbidity and mortality, primarily caused by cardiovascular disease [14,15]. In addition, both short-term and long-term exposure to PM have been associated with the worsening of respiratory conditions and diseases in both adults and children [16,17]. As a function of the inhalation process, PM firstly interacts with the nares mucosa, which is the closest respiratory system (RS) compartment to the external environment, producing a local inflammatory reaction [18]. It has been widely documented, along with RS, that nares harbor a variety of commensal symbiont and pathobiont microorganisms, which, taken together, constitute the nasal microbiota community (NMB) [19,20].

Inhabiting the entire nares surface by niche-specific microorganism, including bacteria, the NMB acts as a gatekeeper to respiratory health, probably impeding respiratory pathogens from setting up an infection [21]. In addition to competitive exclusion function, NMB might also have a role in the anatomical development of the respiratory tract, as well as in the maturation and tolerance of the local immunity [22,23]. However, the continuous exposure of the NMB to large amounts of airborne particles, including PM, can alter the bacterial community composition towards an unstable one that might not be able to resist pathogen overgrowth and to maintain the physiological cross-talk existing with the host, resulting in an alteration of the immune state.

According to these above-mentioned considerations, we recently reported that PM10 and PM2.5 levels of the 3rd day preceding sampling (Day 3) were inversely associated with the majority of analyzed bacterial taxa, except for the *Moraxella* genus. Moreover, two clearly different taxonomical profiles were recognized within the analyzed population, identifying two groups: one characterized by an even community and another widely dominated by the *Moraxella* genus [24]. Therefore, according to the relative abundance of the *Moraxella* genus, we stratified the enrolled subjects into the Mor− and Mor+ groups (the *Moraxella* genus’s cut-off was ≤25% and >25%, respectively), which were characterized by a heterogeneous and an unbalanced NMB, respectively.

The aim of the present study was to evaluate the possible role of NMB in determining plasmatic EV secretion level differences in response to short-term PM exposure levels in a stratified healthy population characterized by two distinct NMB profiles.

## 2. Materials and Methods

Detailed methodological descriptions, including both NMB and EV analyses, as well as subject recruitment, were formerly reported in (Mariani et al., 2017) and (Bonzini et al., 2017; Ferrari et al., 2019), respectively [24,25,26].

Briefly, the involved study population was composed of 51 healthy volunteers recruited between November 2014 and March 2015 by an ad hoc developed announcement posted on the SPHERE Project website (http://users.unimi.it/sphere). In addition, all subjects also filled in a questionnaire collecting exhaustive personal information, including anthropometric characteristics, education, area of residence, job position and location, time spent commuting in traffic, alcohol consumption, smoking habits, drugs, and pre-existing medical conditions.

PM exposure assessment was accomplished using a miniaturized personal sampling device (Personal Cascade Impactor Sampler–PCIS, SKC Inc., PA, USA), retrieving both PM10 and PM2.5 level data during a 24 h period before sample collection. In addition, environmental concentrations of PM10 and PM2.5 of the day preceding the sample collection were also collected from the regional air quality monitoring network (ARPA Lombardia, Milan, Italy) in order to integrate and compare with the ones collected from PCIS.

Each subject underwent a blood drawing and a nasal swab to perform EV and NMB analyses, respectively. Isolation, purification, and characterization of EVs were performed by following the Minimal information for studies of extracellular vesicles (MISEV) 2018 guidelines [27]. Briefly, ethylenediaminetetraacetic acid EDTA-treated blood was centrifuged at 1200× *g* for 15 min at room temperature to obtain platelet-free blood plasma, which was further centrifuged following a three-step centrifugation protocol (1000, 2000, and 3000× *g* for 15 min at 4 °C), and finally ultracentrifuged to obtain an EV-rich pellet (110,000× *g* for 75 min at 4 °C). The EV-rich pellet was then resuspended with 500 µL triple-membrane filtered phosphate-buffered saline PBS, and flow-cytometry (FC) and nanoparticle tracking analyses (NTA) were conducted to assess EV-size distribution, concentration, and EV-origin, using a specific panel of fluorochromes-conjugated antibodies as previously reported, and detailed at http://bit.ly/2sCN9vy [28].

NMB analysis, starting from DNA extraction and amplification from the collected nasal swabs, was carried out through a metabarcoding approach targeting the 16S rRNA V3-V4 hyper-variable regions using the Illumina Miseq sequencing platform, and both upstream and downstream analyses performed on sequencing output were achieved using the default setting suggested in the QIIME 1.9.1 pipeline [24,29]. To confirm the taxonomical composition difference between the two, analyzed group principal coordinate analyses (PCoA) were performed, applying the weighted UniFrac normalized distance metric using QIIME 1.9.1 software.

Plasma was used to quantify tumor necrosis factor alpha (TNF-α) cytokine by the Luminex xMAP^®^-based technology (MYRIAD RBM, Inc., Austin, TX, USA). When the concentration was below the lower limit of quantification (LLOQ), data were replaced by half of the lower limit of quantification (LLOQ/2).

### Statistical Analysis

Descriptive statistics were performed on all variables. Categorical data are presented as frequencies and percentages. Continuous variables are expressed as the mean ± standard deviation (SD) or as the median and interquartile range (Q1–Q3), as appropriate.

Multivariable linear regression models were applied to evaluate the association between EV count and *Moraxella* group (Mor+ vs. Mor−). EV concentrations showed skewed distributions and were naturally log-transformed to achieve normal distribution. For each EV size, we estimated geometric means adjusted for age, gender, smoking habits, and BMI in the Mor+ and Mor− group. Due to the high number of comparisons, we used a multiple comparison method based on Benjamin–Hochberg false discovery rate (FDR) to calculate the FDR P-value. To display the results of the analyses, we used a series graph for EV mean concentrations of each group and a vertical bar chart to represent FDR *p*-values and *p*-values. For the two graphs, the X axis was the size of EVs. The same linear regression model was applied to evaluate the association between TNF-α and total EV.

The role of *Moraxella* relative abundance as a possible modifier of the association between PM exposures and EV count parameters was evaluated, and the multivariable linear regression models were adjusted for age, gender, smoking behavior, and BMI. We observed whether the effect of PM exposure on EV count differed, depending on the *Moraxella* levels stratifying population in two groups (Mor− and Mor+) using 25% of *Moraxella* relative abundance as cut-off, and separate multivariable linear regression models were applied.

Continuous variables were tested for normality and linearity. Then PM exposure data were log-transformed (base 10) to satisfy linearity assumption, and EV counts were log-transformed (base e) to achieve a normal distribution. Effects were thus expressed as geometric mean ratio (GMR) with 95% confidence interval (CI), which corresponds to the exponential of the β regression coefficient when the dependent variable is on the log-scale. GMR indicates the number of times the outcome changes for a 10 times (log10 unit) increase in PM concentration.

Linear regression coefficients in the Mor+ and Mor− subject groups were estimated by this equation:(1)$$\ln\left(EV\right)=\;\alpha +\;\beta_{1}log_{10}\left(PM_{2.5}\right)+\;\beta_{2}AGE+\;\beta_{3}SEX+\;\beta_{4}SMOKE+\;\beta_{5}BMI$$

The whole series interaction was tested by adding interaction term (categorical *Moraxella* * PM) to the multivariable models.

(2)$$\begin{array}{ll} ln\left(EV\right)= & \alpha +\beta_{1}log_{10}\left(PM_{2.5}\right)+\;\beta_{2}Moraxella\left(\pm \right)+\;\beta_{3}log_{10}\left(PM_{2.5}\right)\\ & \;*Moraxella\left(\pm \right)+\beta_{2}AGE+\;\beta_{3}SEX+\;\beta_{4}SMOKE+\;\beta_{5}BMI \end{array}$$

Statistical analyses were performed with SAS software (version 9.4; SAS Institute Inc., Cary, NC, USA).

## 3. Results

### 3.1. Study Population and PM Exposure Data

Among the 51 subjects, a DNA yield sufficient to perform NMB analysis was retrieved for only 40 samples (78%). Interestingly, almost half of the subjects characterized by an insufficient DNA yield were current smokers. The main characteristics of the 40 subjects are listed in Table 1. Participants’ mean age was 48.6 ± 8.4 years, and females represented 57.5% of subjects, while 62.5% of the study subjects were never smokers and 12.5% were classified as current smokers.

PM exposure data measured by PCIS were compared with the data estimated by ARPA monitoring stations (MS). The observed PM10 and PM2.5 median values measured by MS did not diverge from those retrieved by PCIS (PM10: 46.8 µg/m^3^ vs. 45.1 µg/m^3^, respectively; PM2.5: 34.0 µg/m^3^ vs. 36.1 µg/m^3^, respectively; Wilcoxon test for equality of medians: *p* = 0.28 for PM10 and *p* = 0.48 for PM2.5). Moreover, the correlation between the two different PM collection sources was assessed and the results of the Spearman’s rho test were statistically significant (PM10: Spearman’s *r* = 0.59; PM2.5: *r* = 0.68; both *p* < 0.001). The PCoA showed that the enrolled population was clustered into two distinct groups based on its NMB taxonomical composition, defining the Mor− and the Mor+ groups, especially based on PC1 scores when the weighted UniFrac normalized distance metric was applied (Figure 1). In addition, the dissimilarity between the analyzed group was statistically supported by an ANOSIM R value of 0.83 (*p* = 0.001)

As we considered EV size characterized by NTA, the mean EV size values were 224.88 and 215.6 nm for Mor− and Mor+ groups, respectively (*p*-value for differences >0.1). Modal EV size values span from 169.9 nm for the Mor− group and 157.36 nm for the Mor+ one.

As we are aware that the information on mean and mode might not be exhaustive, we further compared the two groups in terms of distribution of mean vesicle concentrations for each size (Figure 2). In the upper part of the figure, we reported for each EV size (from 30 to 700 nm) the mean concentration calculated in each group. The Mor+ and Mor− subjects’ size distributions were similar, as confirmed by the lower part of the plot, which reports the *p*-values and FDR *p*-values obtained comparing Mor+ vs. Mor−.

### 3.2. Association between Plasmatic EV Concentration and Inflammatory Markers in the Whole Population

In order to verify the effect of plasmatic EV concentration in modifying the inflammatory state, multivariable linear regression analysis between total EV count and TNF-α, as an example of a pro-inflammatory cytokine, adjusted for age, sex, smoking behavior, and BMI, was performed. The total count of plasmatic EVs was positively associated with the plasmatic concentration of TNF-α (GMR = 1.035, *p* = 0.023) (Figure 3).

### 3.3. PM Exposure Effects on Plasmatic EV Levels in the Stratified Population (Mor− and Mor+ Groups)

To assess whether and how an unbalanced NMB profile may modify the response to PM10 and PM2.5 exposure in terms of plasmatic EV count, the 40 subjects were stratified, according to *Moraxella* genus relative abundance, into Mor− (*n* = 30) and Mor+ (*n* = 10) groups (*Moraxella* genus relative abundance ≤25% and >25%, respectively).

Multivariable analyses computed on both FC and NTA outcomes showed (Figure 4), among Mor− subjects, positive associations between PM2.5 exposure and both endothelial-derived (CD105+) (GMR = 4.39 *p* = 0.02) and EV total count (GMR = 1.92; *p* = 0.02). By contrast, negative associations were identified for the Mor+ group between PM2.5 exposure and neutrophil-derived EVs (CD66+) (GMR = 0.004; *p* = 0.01) and between PM2.5 exposure and epithelium-derived EVs (EpCAM+) (GMR = 0.005; *p* = 0.01).

Both groups exhibited a significant––albeit opposite––association between PM2.5 exposure levels and monocyte-derived EVs (CD14+) (Mor−: GMR = 5.34; *p* = 0.04/Mor+: GMR = 0.04; *p* = 0.02).

Focusing on platelet-derived EVs (CD61+), both the Mor− and Mor+ groups followed the previously mentioned EV association direction when PM2.5 exposure was considered, even though no statistical associations were clearly identified. We repeated our analysis for PM10 exposure effects and observed consistent trends for the previously described outcomes (except for EpCAM+ EVs, which did not present any clear association with PM10 exposure) (Appendix A).

## 4. Discussion

In this study, we investigated the effects of short-term PM exposure on plasmatic EV levels in a healthy population, stratified into Mor− and Mor+ groups according to NMB *Moraxella* genus relative abundance, in order to assess the possible role played by the nasal bacteria community as a factor of susceptibility or resistance to the widely known adverse-inhaled pollutant effects.

In one study, we showed that exposure to PM can affect the microbiota community [24]. These changes lead to modifications of the indigenous bacterial community, perturbing the structure and the relationships existing between the different species of the microbiota and potentially modifying the equilibrium between the bacteria community and the host, which often leads to unhealthy conditions [30,31]. In addition, PM exposure can trigger inflammatory response [32], worsen chronic conditions spread all over the body sites, and also heighten the risk for acute and chronic diseases [33,34,35,36,37,38]. Several epidemiological studies showed how the effects exerted by PM exposure on human health can be attributed, at least in part, to plasmatic EV modifications, which include variations in the transcriptomic and proteomic content, as well as in the circulating amount of different EV types [39,40,41,42,43]. In addition, documentation shows that PM exposure may trigger EV release in a dose-dependent manner [41], inducing the release of proinflammatory cytokines such as IL6 and TNF-α [44].

In agreement with this evidence, we observed a positive association between the total plasmatic EV count and TNF-α level, considering the whole enrolled population. This observed increment of TNF-α could be caused by a partially enhanced EV activity in response to the acute inflammatory response induced by PM exposure [44].

In the Mor− group, the positive associations between both PM10 and PM2.5 levels and plasmatic EV could be partially explained by the presence of a heterogeneous NMB. Interestingly, both CD14+ and CD105+ EVs were the most abundant ones, probably due to the fact that, after deposition on alveolar epithelium, PM can be phagocyted by macrophages, pass through the alveolar–capillary membrane, and directly interact with pulmonary endothelium [45], stimulating these sources of EVs production.

By contrast, Mor+ subjects showed a negative association between measured PM10 and PM2.5 exposures and EV derived from neutrophils, epithelium, and macrophages.

Despite the uncertain role exerted by EVs in the inflammatory processes, such as after PM exposure, an increasing number of studies highlight the possible mitigative role of the above- mentioned EVs during inflammation [46]. In particular, CD105+ EVs could probably promote vascular regeneration, either through a specific interaction or through miRNA delivering into recipient cells. Indeed, it has been found that CD105+ EVs-carried miRNA-222 may contribute to weaken inflammatory effects modulating the endothelial expression of ICAM-1 [47], as well as for the PCSK9 protein, which has been related as a biomarker for different cardiovascular negative conditions [48].

An inflammatory regulating behavior has been also identified for CD14+ EVs that in the respiratory tract may contribute to control both cytokine signaling and IFNγ-induced activation of signal transducer and activator of transcription (STAT), which is responsible for enhancing the expression of proinflammatory genes (STAT-dependent genes) [49].

Similarly to the above-mentioned evidence, one report determined that small EVs might be able to prevent macrophages activation through the M1-proinflammatory phenotype after stimulation with LPS, in both in vitro and in vivo models, suggesting a possible anti-inflammatory role exerted by EVs [50].

Together, these findings suggest that the group with a balanced nasal bacteria community seems to have a more reactive response to PM insults, in contrast to the Mor+ group, considering the amount of plasmatic EVs. This different behavior identified for the Mor+ subjects could be addressed to the unbalanced *Moraxella*-dominated NMB, which might modify the protective function provided to the host against inhaled pollutant effects.

In contrast with these considerations, it was observed that PM exposure was linked to an increased concentration of phosphatidylserine-enriched EVs released from different cell types. These EVs alter the phagocytes’ efferocytotic activity [41], which has been linked to the worsening of pathological conditions such as atherosclerosis and chronic obstructive pulmonary disease [51,52]. In addition, through biological pathway analysis, PM exposure seems to modify EV-encapsulated miRNA involved in the maintenance of health [12,42,53].

Thus, merging the aforementioned and our previous results [24], the nasal bacterial community could be considered as the first compartment targeted by PM exposure inflammation, as well as a sort of filter between the host and the external environment, which might alter the peripheral effects exerted by air pollution exposure in terms of plasmatic EV levels. This study has limitations. First, we were not able to discriminate through FC the EV belonging to the different bacterial strains, which could be affected by PM exposure in terms of plasmatic concentration and also involved in the immune state regulation. Second, we considered only NMB, although other microbiota compartments, such as the gastrointestinal one, could be modified by PM exposure and contribute to the total EV cargo circulating through the human body. However, since the primary aim of this study was to assess how the microbiota modifies the variation of plasmatic EVs induced by short-term PM exposure, we focused our attention on the nasal bacterial community, considering it as the first target of PM exposure.

## 5. Conclusions

To the best of our knowledge, this is the first study specifically addressing the role of NMB in determining differences in plasmatic EV secretion levels in response to short-term PM exposure levels in healthy subjects.

Future studies will be carried out on a larger population, with the purpose of a deeper characterization of the NMB components, understating the relationship between the host and the nasal bacteria community and how it could react to air pollution exposure. A focused insight into EV contents will be performed in order to clarify if a different plasmatic concentration of EVs could be also linked to different transported biomolecules, as well as the characterization of the bacterial secreted EVs, which taken together may be informative of physiological changes exerted by PM exposure.

## Figures and Tables

**Figure 1 ijerph-17-00611-f001:**
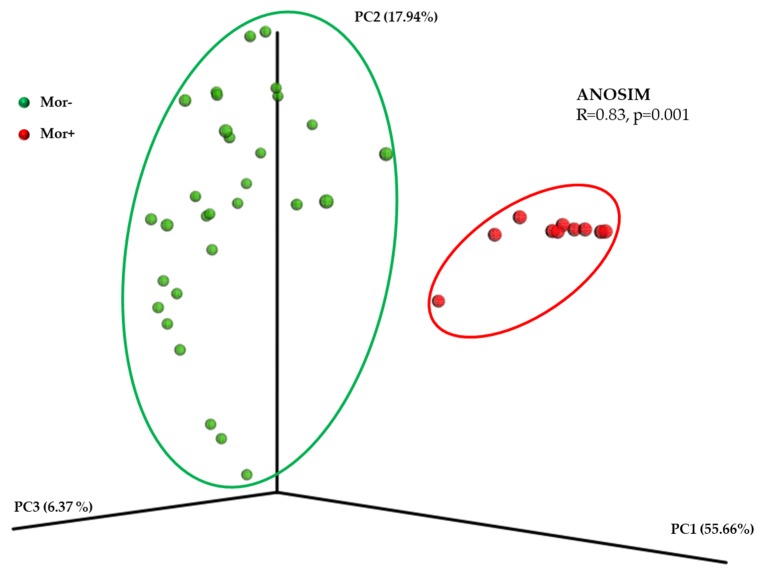
Principal coordinate analyses (PCoA) plot made using the normalized weighted UniFrac distance metric. Each dot corresponds to a single subject belonging either to Mor− (green dot) or Mor+ (red dot). The variance explained by each axis is given in parentheses. Dissimilarity between group was statistically tested applying the ANOSIM method.

**Figure 2 ijerph-17-00611-f002:**
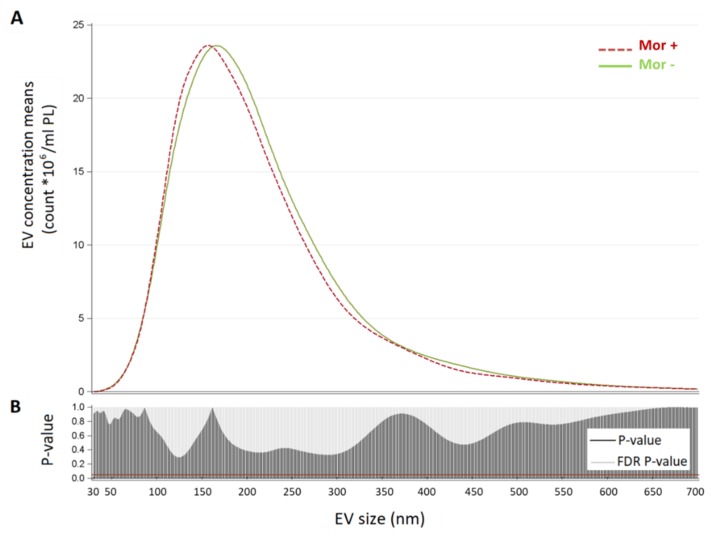
Extracellular vesicle (EV) size distribution in the Mor− and Mor+ groups. Panel (**A**): * Reported geometric means were adjusted for age, sex, BMI, and smoking habits. Plots showing for each group (Mor− and Mor+) the distribution of mean vesicle concentrations for each size. Panel (**B**): vertical bar charts represent FDR and *p*-value for each size comparison; the red line indicates *p*-value = 0.05.

**Figure 3 ijerph-17-00611-f003:**
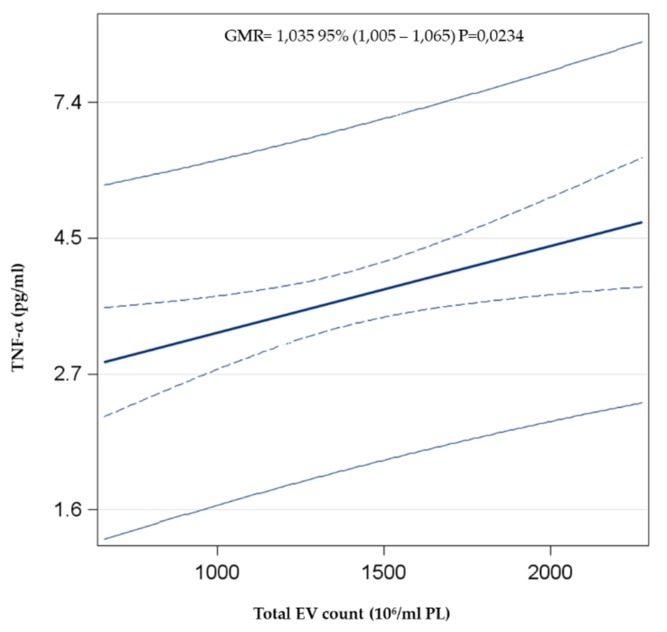
Association between total EVs and TNF-α. The multivariable linear regression model was adjusted for age, gender, smoking behavior (Never smoker, Former smoker, Current smoker), and BMI.

**Figure 4 ijerph-17-00611-f004:**
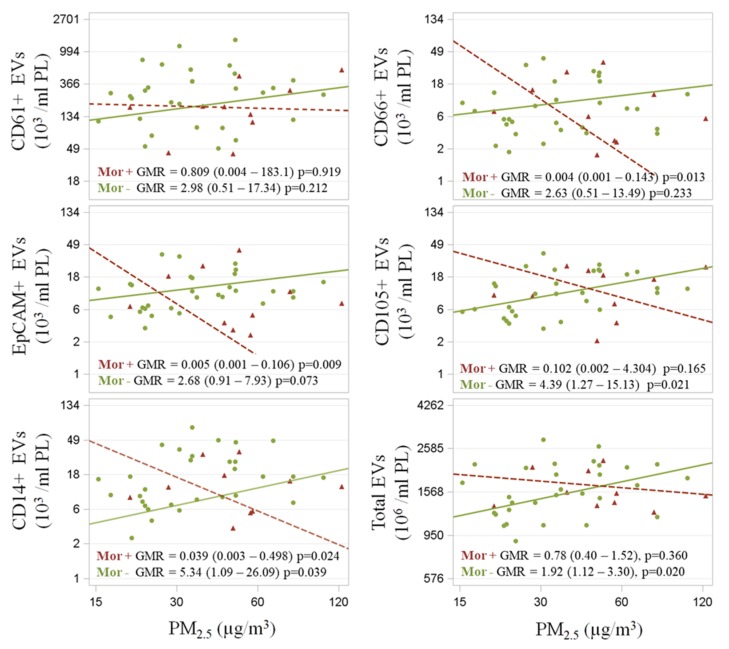
Association between EV outcomes and PM2.5 exposure. Mor+ and Mor− subjects were represented by crosses and circles, respectively. Scatterplots of EV (103/mL PL) vs. PM2.5 (below) levels (µg/m^3^). Covariate-adjusted geometric mean ratios and corresponding 95% confidence intervals (GMR (95%CI)) in EV estimated per log10-unit increase in PM are shown. Subjects were stratified according to their *Moraxella* genus relative abundance into the Mor− (≤25%) and Mor+ (>25%) group. Total EV count obtained via nanoparticle tracking analysis (NTA). EV fraction counts performed via flow-cytometry (FC) analysis.

**Table 1 ijerph-17-00611-t001:** Characteristics of the study participants in the Mor− and Mor+ groups.

Characteristics	Mor−	Mor+
	(*n* = 30)	(*n* = 10)
*Moraxella*, *Relative Abundance (%)*	0.2 [0.04; 0.5]	90.2 [84.4; 93.8]
**Age,** *Years*	48.6 ± 8.5	48.4 ± 8.4
**Sex,***n* (%)		
*Male*	13 (43.3%)	4 (40%)
*Female*	17 (56.7%)	6 (60%)
**BMI,** *Kg/m^2^*	24.6 ± 3.2	24.6 ± 3.4
*<25*	16 (53.3%)	5 (50.0%)
*≥25*	14 (46.7%)	5 (50.0%)
**Smoking habits,***n* (%)		
*Never Smoker*	17 (56.7%)	8 (80%)
*Former Smoker*	10 (33.3%)	0 (0.0%)
*Current Smoker*	3 (10%)	2 (20%)
**Education,***n* (%)		
*High School*	8 (26.7%)	4 (40%)
*University*	18 (60%)	6 (60%)
*Others*	4 (13.3%)	0 (0.0%)
**Living Area,***n* (%)		
*City of Milan*	14 (46.7%)	3 (30%)
*Province of Milan, Outside City Area*	9 (30%)	3 (30%)
*Province of Monza-Brianza*	4 (13.3%)	1 (10%)
*Other Provinces in Lombardy*	3 (10%)	3 (30%)

Continuous variables are expressed as mean ± standard deviation (SD) or as median (first quartile–third quartile) if not normally distributed; discrete variables are expressed as counts (%).

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
