# Peer review of "Nasal Microbiota Modifies the Effects of Particulate Air Pollution on Plasma Extracellular Vesicles"

_ijerph, 2020, doi:10.3390/ijerph17020611_

Round 1
Reviewer 1 Report
The article by Mariani et al. investigated the effect of particulate matter exposure on plasmatic EV levels in two different populations: one with a balanced nasal microbiota (Mor-) vs. another with an unbalanced nasal microbiota (Mor+). This report is the logical continuation of previous studies from the research group. While the results are interesting, some aspects of the manuscript are preliminary. More data are needed to merit publication.
Comments:
The authors have classified the subjects into two different populations based on their nasal microbiota. However, differences in the microbiota from the gastrointestinal tract may also affect the total plasmatic EV number. Can the authors discuss this?
The authors have used NTA analysis to determine the concentration of EVs. This technology also determines the size distribution of the particles, and these data should be included in the manuscript. Ultracentrifugation isolates different types of EVs. What is the mean size of the EV population? Have the authors found differences in the size distribution of EVs between groups?
Among the subtypes of EVs, exosomes are particularly important as they contain regulatory molecules that could be transferred into recipient cells to regulate their function. The EV population obtained in this study was not characterized. It would be interesting to see the expression of exosomal markers such as CD9, CD63, and CD81. FC can do this in a similar way as the authors have done for other surface markers.
Discussion section (line 162). Why there is a different response in terms of the association between total plasmatic EVs and exposure to particulate matter in the Mor- vs. Mor+ groups is not clear. The authors speculate about the potential role of a balanced vs. unbalanced nasal microbiota. What is the role of the secreted EVs after exposure to particulate matter? The authors suggest that these EVs may have an anti-inflammatory activity (this can be tested, see the following point). The manuscript can gain much interest if some of these questions can be addressed.
Discussion section (line 183). Regarding the role of EVs in inflammation, a recent report by Pacienza et al. (Mol Ther Methods Clin Dev. 2018; 13:67-76) demonstrated that EVs could prevent the activation of macrophages to the M1-proinflammatory phenotype after stimulation with LPS.
Author Response
REVIEWER 1
COMMENT 1: The authors have classified the subjects into two different populations based on their nasal microbiota. However, differences in the microbiota from the gastrointestinal tract may also affect the total plasmatic EV number. Can the authors discuss this?
We thank the Reviewer for this suggestion. We agree that PM exposure could alter also gastrointestinal microbiota, such as the microbiota inhabiting other body sites, and the whole body microbiota could take part in modifying the human plasmatic EV concentration. However, since nasal microbiota is considered as the gatekeeper of respiratory health and therefore the main target of PM exposure, we focused our study on the nasal bacterial community. However, we now acknowledged this limitation in the discussion section (page 9, lines 271).
COMMENT 2: The authors have used NTA analysis to determine the concentration of EVs. This technology also determines the size distribution of the particles, and these data should be included in the manuscript. Ultracentrifugation isolates different types of EVs. What is the mean size of the EV population? Have the authors found differences in the size distribution of EVs between groups?
In the revised version of the manuscript, we extensively described the size distribution of EVs, as measured by NTA.
First, we added in the result section the information regarding the mean and the modal EV size (page 5, lines 156).
Second, we reported as figure 2 the size distribution observed in the two groups (Mor+ vs Mor-).
COMMENT 3: Among the subtypes of EVs, exosomes are particularly important as they contain regulatory molecules that could be transferred into recipient cells to regulate their function. The EV population obtained in this study was not characterized. It would be interesting to see the expression of exosomal markers such as CD9, CD63, and CD81. FC can do this in a similar way as the authors have done for other surface markers.
Unfortunately, since we performed EV characterization only on fresh EVs in order to maximize the quality of the analysis, we are not able to analyse the proposed additional marker on the stored samples. Moreover, as stated in the MISEV 2018 guidelines ( https://doi.org/10.1080/20013078. 2018.1535750) the abovementioned markers have been recently considered as nonspecific markers for exosomes.
COMMENT 4: Discussion section (line 162). Why there is a different response in terms of the association between total plasmatic EVs and exposure to particulate matter in the Mor- vs. Mor+ groups is not clear. The authors speculate about the potential role of a balanced vs. unbalanced nasal microbiota. What is the role of the secreted EVs after exposure to particulate matter? The authors suggest that these EVs may have an anti-inflammatory activity (this can be tested, see the following point). The manuscript can gain much interest if some of these questions can be addressed. Discussion section (line 183).
The question raised by reviewer 1 is very important, but very difficult to assess in an epidemiological study.
In the attempt to partially answer to this question, we measured TNF-alpha, as a paradigm of a pro-inflammatory cytokine, and evaluated the association with the total count of EVs. We found a positive significant association which has been reported as figure 3.
COMMENT 5: Regarding the role of EVs in inflammation, a recent report by Pacienza et al. (Mol Ther Methods Clin Dev. 2018; 13:67-76) demonstrated that EVs could prevent the activation of macrophages to the M1-proinflammatory phenotype after stimulation with LPS.
We added and discussed the reference in the discussion section (page 9, line 249)
Reviewer 2 Report
The article entitled “Nasal Microbiota Modifies the Effects of Particulate Air Pollution on Plasma Extracellular Vesicles” is an interesting effort to relate air pollutants, microbiota, and circulating EVs. My major concern about the article is that a complex multivariate study has been applied with a lower number of samples, reporting just a change in the sign of the coefficient for a non-validate model. I think the reader will gain interest if more clear presentation of the data, even if not significant, where reported, and finalize with the complete presentation of the model, including the changes in the trends observed between Mor- and Mor+ indivuals.
Some comments;
In the introduction, document better the rationale behind the proposed aim. Why different flora may affect the response in terms of circulating EVs to air pollution exposure? Better description of the microbiota profiling. How abundance of different species is assessed? Complete report of microbiota profiling. Presentation of the data by PCA… are the data cluster somehow? The term exosomes is mentioned on the Supplementary Table I, as a fraction of the NTA. But no mention of which fraction that would be in the M&M section. My guess is that it should be removed from supplementary data? The model proposed predict, based on different variables, the amount of circulating vesicles. Is the model significant? Could be the model be validated somehow –by bootstrapping for instance? Report completely the model, to know if the opposite sign of GMRs reported for microbiota population, is also observed for other variables, such sex, BMI or smoking history? That would give the idea of possible confounding factors hiding in the results. The discussion is unclear. Circulating EVs could have an involvement in inflammation, so balance microbiota favor a better anti-inflammatory response? In certain passages of the discussion this seems to be the argument, but also the authors mention the opposite (a more acute response to pollutants). Were some markers of inflammations analyzed in the blood samples?Author Response
Please see the attachment

Round 2
Reviewer 1 Report
The article by Mariani et al has greatly improved. The authors have answered all the concerns raised in the revision. In addition, new data further support the conclusions. I believe the manuscript now merits publication in IJERPH.
Reviewer 2 Report
The authors had tried to answer all the questions raised and they had modified accordingly the manuscript. I think the manuscript is ready for publication. Just few minor comments;
a) the article Mariani et al 2017 is not in the reference list (there is a Mariani et al 2018?)
b) I think it would be interesting to report in the supplementary information the eigenvectors or display the loading plot of the PCA analysis
c) Could you show the actual points that set up the correlation on the figure 3?